

# Young children do not require perceptual-motor feedback to solve Aesop's Fable tasks

Rachael Miller[1,*], Sarah A. Jelbert[1,*], Elsa Loissel[1], Alex H. Taylor[2] and Nicola S. Clayton[1]

[1] Department of Psychology, University of Cambridge, Cambridge, United Kingdom
[2] School of Psychology, University of Auckland, Auckland, New Zealand
[*] These authors contributed equally to this work.

Corresponding author
Sarah A. Jelbert, saj48@cam.ac.uk

## ABSTRACT

Aesop's Fable tasks—in which subjects drop objects into a water-filled tube to raise the water level and obtain out-of-reach floating rewards —have been used to test for causal understanding of water displacement in both young children and non-human animals. However, a number of alternative explanations for success on these tasks have yet to be ruled out. One hypothesis is that subjects may respond to perceptual-motor feedback: repeating those actions that bring the reward incrementally closer. Here, we devised a novel, forced-choice version of the Aesop's Fable task to assess whether subjects can solve water displacement tasks when this type of feedback is removed. Subjects had to select only one set of objects, or one type of tube, into which all objects were dropped at once, and the effect the objects had on the water level was visually concealed. In the current experiment, fifty-five 5–9 year old children were tested in six different conditions in which we either varied object properties (floating vs. sinking, hollow vs. solid, large vs. small and too large vs. small objects), the water level (high vs. low) and/or the tube size (narrow vs. wide). We found that children aged 8–9 years old were able to solve most of the water displacement tasks on their first trial, without any opportunity for feedback, suggesting that they mentally simulated the results of their actions before making a choice. Children aged 5–7 years solved two conditions on their first trial (large vs. small objects and high- vs. low-water levels), and learnt to solve most of the remaining conditions over five trials. The developmental pattern shown here is comparable to previous studies using the standard Aesop's Fable task, where eight year olds are typically successful from their first trial and 5–7 year olds learn to pass over five trials. Thus, our results indicate that children do not depend on perceptual-motor feedback to solve these water displacement tasks. The forced-choice paradigm we describe could be used comparatively to test whether or not non-human animals require visual feedback to solve water displacement tasks.

## INTRODUCTION

Recently, a number of comparative, non-linguistic studies have been conducted to determine what young children and non-human animals understand about elements of water displacement. Researchers using the floating peanut task have demonstrated that children and some great apes will spontaneously pour or spit water into a tube in order to bring a floating peanut within reach (*Mendes, Hanus & Call, 2007*; *Hanus et al., 2011*). Among children, only a small proportion of four year olds, but up to 75% of 8 year olds, recognise that they can use water as a tool to raise the level of floating rewards (*Hanus et al., 2011*). A related line of research has used the Aesop's Fable paradigm to assess whether children and non-human animals possess a causal understanding of water displacement (*Bird & Emery, 2009*; *Cheke, Bird & Clayton, 2011*; *Taylor et al., 2011*; *Clayton, 2014*; *Jelbert, Taylor & Gray, 2015*). These tasks are analogous to Aesop's famous tale in which a thirsty crow drops stones into a pitcher of water to raise the water level until it is high enough for the bird to drink, except for the fact that the subjects in the experiments are not thirsty but a reward is placed on top of the water. In Aesop's Fable tasks, subjects are typically presented with a choice of objects to drop into tubes, or a choice of tubes to drop objects into, where one option is the most (or only) functional choice to raise the water level and obtain a floating out-of-reach reward.

When presented with versions of this task, 4–7 year old children appear to learn, over the course of five trials, which options will allow them to obtain the reward (*Cheke, Loissel & Clayton, 2012*). Across three conditions, 4–7 year olds could learn to drop stones into a tube containing water, rather than one containing sawdust, and 5–7 year olds could learn to drop objects that sank, rather than objects that floated on the water's surface. The majority of children aged seven and over, but few younger children, also learnt to pass a task including counter-intuitive causal cues, where dropping a stone into one tube also raised the water level in a second tube via a concealed connection (*Cheke, Loissel & Clayton, 2012*). More recently, 5–7 year old children failed to pass a more difficult version of this task involving solid and hollow objects within five trials, but learnt to do so within 20 trials (*Miller et al., 2016*).

Strikingly, performance on Aesop's Fable tasks by some bird species—primarily corvids, but also, to some degree, grackles (*Quiscalus mexicanus*)—has been shown to rival that of 5–7 year old children (*Bird & Emery, 2009*; *Cheke, Bird & Clayton, 2011*; *Taylor et al., 2011*; *Logan, 2015*; *Logan, 2016*). Rooks (*Corvus frugilegus*), Eurasian jays (*Garrulus glandarius*) and New Caledonian crows (*Corvus moneduloides*), for example, have all been tested on various Aesop's Fable tasks. These experiments revealed that corvids will drop sinking rather than floating objects into water-filled tubes, will drop large rather than small objects, and drop solid objects (that displace a large amount of water) rather than hollow objects (that displace only a small amount). They preferentially drop objects into tubes containing water, rather than tubes containing sand, and drop objects into tubes with a high- rather than a low-water level (rooks: *Bird & Emery, 2009*; Eurasian jays: *Cheke, Bird & Clayton, 2011*; New Caledonian crows: *Taylor et al., 2011*; *Jelbert et al., 2014*; *Logan et al., 2014*). In most of these cases, birds do not solve the task on their very first trial, but they do learn to
solve the tasks over a small number of trials, rapidly learning to exclusively select the most (or only) functional option. Thus, their behaviour is highly similar to that of 5–7 year old children. Only by the age of eight years do children reliably choose correct options on their first trial, at which point children's performance clearly differs from that of corvids (*Cheke, Loissel & Clayton, 2012*).

Although birds and 5–7 year old children show similar learning patterns on the Aesop's Fable task, to date, it remains unclear whether their comparable performance is underpinned by similar cognitive mechanisms (see *Clayton, 2014*; *Jelbert, Taylor & Gray, 2015* for review). Success on the Aesop's Fable tasks could be achieved through using a causal understanding of water displacement and an ability to mentally simulate the effect that dropping objects will have on the water level in each tube. However, the pattern of learning shown by birds and 5–7 year old children could also be explained by other mechanisms. A common feature of the Aesop's Fable tasks is that the reward incrementally moves closer to the subject's reach with each stone that is dropped into the tube to raise the water level. Therefore, subjects could learn to solve these tasks by repeating those actions that bring the reward incrementally closer—i.e., by responding to perceptual-motor feedback (*Taylor & Gray, 2009*; *Cheke, Bird & Clayton, 2011*; *Jelbert, Taylor & Gray, 2015*). This type of feedback is thought to underpin the seemingly 'insightful' behaviour by which birds spontaneously learn to pull up strings to bring in attached rewards (New Caledonian crows: *Taylor et al., 2010*; *Taylor, Knaebe & Gray, 2012*; common ravens, *Corvus corax*: *Heinrich & Bugnyar, 2005*; California scrub jays, *Aphelocoma californica*: *Hofmann, Cheke & Clayton, 2016*), and is a plausible explanation for the birds' behaviour on water displacement tasks. For example, *Cheke, Bird & Clayton (2011)* found that one out of two Eurasian jays could pass an arbitrary task where a reward was pushed incrementally towards the subject each time they dropped a stone into an L-shaped apparatus. However, they failed to learn a task with the same reward schedule where the subject was given a reward by the experimenter once they had dropped a certain number of stones into one of two coloured tubes. This suggests that, in some problem-solving situations, corvids potentially learn by attending to the position of a reward after each action they make.

It is currently unclear whether responding to perceptual-motor feedback contributes to children's performance on Aesop's Fable tasks. The increase in first trial success that occurs between seven and eight years is roughly in line with performance on classic Piagetian conservation of volume tasks, which are typically passed around age seven (*Piaget, 1930*; *Piaget, 1974*). However, Cheke and colleagues found that performance on one conservation of volume task—in which water was poured from a short, wide container into a thin, narrow container and children were asked whether the amount of water was now more, less or the same—did not predict children's performance on various Aesop's Fable tasks (*Cheke, Loissel & Clayton, 2012*). This suggests that some children might pass these water displacement tasks by attending to covariation cues, rather than by using an understanding of water displacement. With this issue in mind, in the current study we devised a novel forced-choice version of the Aesop's Fable paradigm, capable of determining whether subjects can solve Aesop's Fable tasks when opportunities for perceptual-motor

feedback are removed. This task was designed to be appropriate for use with both human and non-human populations.

In our forced-choice paradigm, children were presented with two versions of a modified water-tube apparatus, where a sliding barrier could be pushed to release a set of pre-positioned objects, all at once, into a water-filled tube. On each trial, either the two water-tubes or the two sets of objects varied, and children could choose one apparatus to interact with only. Tubes were transparent on one side and opaque on the other. Thus, after the child had indicated their tube of choice, but before they slid the barrier to release the objects, the tube could be rotated to the opaque side, which denied the child visual access to the water level rising when the objects dropped into the tube. Because of these design differences, (in that one set of objects were dropped in a tube, all at once, and the effect on the water level was concealed) here subjects could not succeed by observing the effect that dropping each object had on the water level, as may have been the case in previous studies.

Subjects received five trials in each condition, which allowed us to address two questions. The first was whether participants could solve these tasks on their very first trial. First trial success would indicate that the participant could reason causally—likely mentally simulating the effects that objects would have on the water level—before they received any kind of feedback from their actions. The second question was whether children could learn to solve the tasks over a small number of trials, despite not witnessing the water level rising in response to objects being dropped into the tube during the trials. At the end of each trial, children received feedback on the overall success of their actions (whether the reward could now be reached), but they did not receive the specific type of perceptual-motor feedback of observing the reward move incrementally closer, that has been suggested as an explanation for success on these types of water displacement task (*Taylor & Gray, 2009*; *Cheke, Bird & Clayton, 2011*; *Jelbert, Taylor & Gray, 2015*).

Across six conditions, we either varied the properties of the objects (small vs. large objects, too large vs. small objects, sinking vs. floating objects and hollow vs. solid objects) or the properties of the tubes that were presented (narrow vs. wide tubes, and high vs. low water levels) to assess what children understand about water displacement. Two conditions had been used previously with young children (sinking vs. floating & solid vs. hollow objects: *Cheke, Loissel & Clayton, 2012*; *Miller et al., 2016*) and four had not. Five of six tasks (all except the too large vs. small objects condition) had been used in previous studies with corvids (*Taylor et al., 2011*; *Jelbert et al., 2014*; *Logan et al., 2014*). Where possible, the conditions that we presented were designed to counterbalance each other, so that the functional choice in one condition was non-functional in another condition, and therefore a preference for one particular object or tube could be ruled out (*Jelbert, Taylor & Gray, 2015*). For example, in the narrow vs. wide tube condition, only the narrow tube was functional, whereas in the high- vs. low-water level condition, only the wide tube was functional. If children's success is indeed dependent on perceptual-motor feedback, we expected children to perform more poorly on the current forced-choice tasks than they did in previous studies where perceptual-motor feedback was freely available.

## METHODS

### Subjects

Subjects were 55 children aged between five and nine years old: 10 5-year olds (Mean: 5.4 years; Range: 5.0–5.9 years), 13 6-year olds (M: 6.4; R: 6.0–6.9), 11 7-year olds (M: 7.6; R: 7.2–7.9), 11 8-year olds (M: 8.4; R: 8.0–8.9) and 10 9-year olds (M: 9.4, R:9.1–9.9), of which 27 were male and 28 were female. This sample size was chosen to ensure we had a minimum of 10 children per age group, and that we included a similar number of participants to the previous Aesop's Fable study with children (*Cheke, Loissel & Clayton, 2012*). All children completed both testing sessions. Children were recruited and tested at five primary schools in Cambridgeshire, serving predominantly white, middle-class communities, between February and May 2016.

### Apparatus

In all experimental trials, subjects were presented with two water-filled Perspex tubes each containing a magnetic floating token, which could be retrieved using a magnetic 'fishing rod' when the water level reached 60 mm from the top of the tube. A removable slanting Perspex tube containing objects (referred to as the 'object-tube') was placed on top of each tube (diameter 5 cm, Fig. 1). Subjects could slide a barrier at the base of the object-tube to release the objects, all at once, into the water-filled tube. Different tubes and different objects were used in each condition, and details of these are provided in the experimental procedures below.

Tokens were small pieces of cork attached to a small magnet ($\sim$5–10 mm$^2$), which would float on the water's surface. The fishing rod comprised 60 mm of string with a small magnet attached to one end and a sheet of clear plastic ($\sim$90 $\times$ 60 mm) at the other, which prevented the rod from being inserted fully into the tube. Using a fishing rod ensured that the token was accessible at the same level, across all conditions and for all children. To maintain motivation, we used a sticker reward trail where subjects would receive a sticker after every few correct choices. Each time a subject retrieved a token, they could move a playing piece one step on this trail and stickers were placed intermittently across the trail (approximately every 3rd step) for the child to obtain.

### Pre-Training

Subjects received two training steps, first to learn to slide the barriers attached to the object-tube, and second to learn that only one option was rewarded, and that only one choice was permitted. In the first training step, subjects were presented with a Perspex collapsible platform apparatus (as per *Bird & Emery, 2009*), and observed the experimenter placing the slanting object-tube on top of the apparatus. The experimenter demonstrated inserting a training object (a plastic, light blue oblong, 25 $\times$ 15 $\times$ 10 mm) into the object-tube. Then, the experimenter pushed the barrier to allow the object to drop down and collapse the platform inside the apparatus to release a token. We used this opportunity to explain that the tokens were equivalent to one step along the sticker reward trail and, if they reached a sticker, then the sticker belonged to the subject. The subjects were then asked to drop the

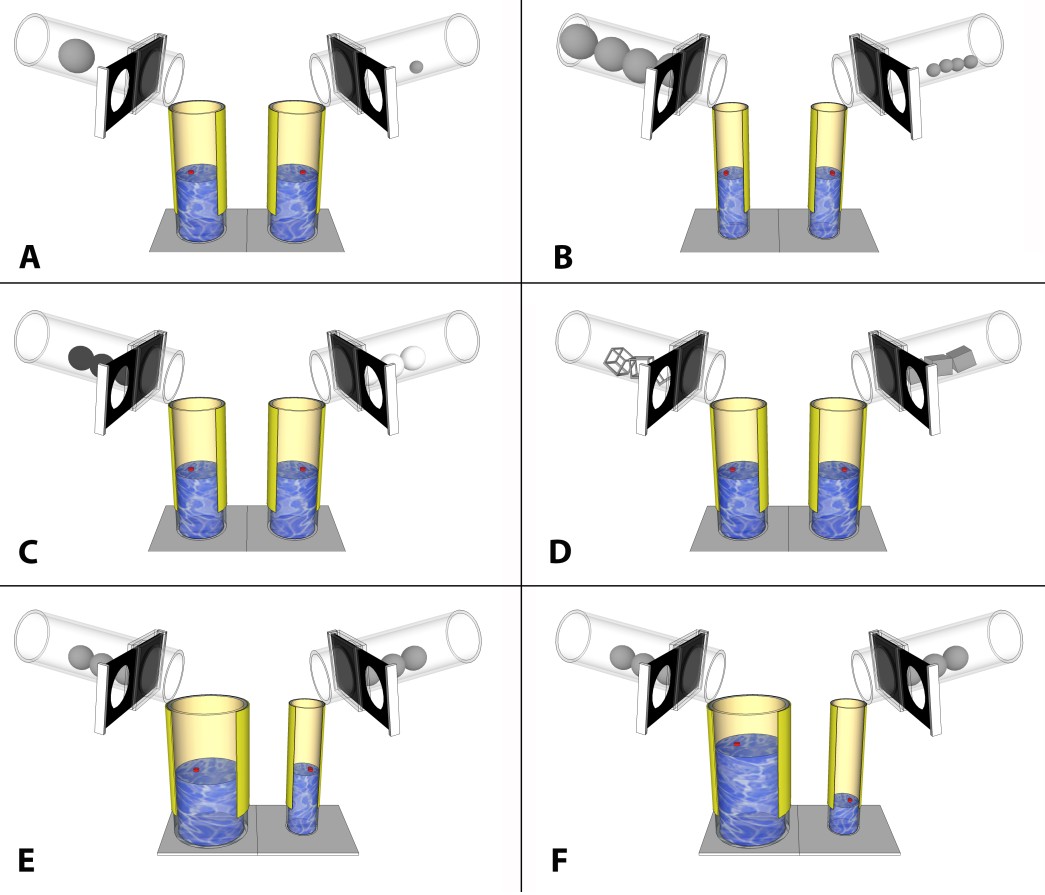

**Figure 1** **Diagram of each experimental condition.** Children were first presented with two tubes in the orientations displayed here. After selecting one tube, the experimenter removed the non-chosen tube, and rotated the chosen water-filled tube 180° to obscure the child's view of the water level with the yellow covering. The child could then slide the barrier to release objects into the tube. (A) large vs. small, (B) too large vs. small, (C) floating vs. sinking, (D) hollow vs. solid, (E) wide vs. narrow tubes, (F) high vs. low water levels in wide vs. narrow tubes.

blue training object into the collapsible platform apparatus themselves, and show that they were able to push the barrier to release the object.

In the second training step, subjects received five trials of a pebbles vs. water condition (analogous to earlier sawdust vs. water conditions with birds). We presented the subject with two medium vertical Perspex tubes (as used in the main tests) with the object-tube attachment in position and demonstrated inserting three of the blue oblong training objects into the object-tubes. One vertical tube was filled with small blue pebbles, and the other tube was filled with water to the same height. At the start of the test the level of both the water and the pebbles was too low for the child to reach the token with the magnetic fishing rod. Releasing objects into the water-filled tube only would raise the water level sufficiently to bring the token within reach. Here, we used small blue pebbles in place of sawdust in order to avoid any potential allergies to this substrate in the children. Subjects completed five trials, or three correct consecutive trials, where they could choose only one tube by

pushing the barrier, which released all of the objects into the vertical tube. They could then use the fishing rod to attempt to obtain the token. This step allowed us to ensure the subject understood that they could only make one choice of tube and that there was only one correct choice (in this case the water-filled tube). They could also practice using the fishing rod to obtain the token.

## Test conditions

Immediately after training, subjects began the experimental trials. They received 30 trials in total (six conditions, five trials per condition), generally completing 18 trials in session one (which lasted 30–45 min), and 12 trials in a second session the following day (lasting 20–30 min). Condition order was pseudorandomised (see details in Test Procedure), with trials from each condition spanning both testing days. In each condition, a different set of tubes or objects were used (see Supplemental Information 1).

**Condition 1: Large vs. small objects.** Two identical medium sized water-filled tubes (diameter = 5 cm, height = 15 cm) were presented, with the water set to the same level. One tube was presented with a single large object (grey clay sphere: 40 mm diameter), and one tube with a single small object (grey clay sphere: 13 mm diameter). When released, the large object raised the water level sufficiently to bring the token within reach, but the small object did not.

**Condition 2: Too large vs. small objects in narrow tubes.** In the second condition, four large and four small objects were presented, equivalent to those used in condition 1. Here, two identical *narrow* water-filled tubes were used (diameter = 3.5 cm, height = 15 cm). The water level was equivalent in both. Because narrower tubes were used, the large object was now too large to fit inside the water-filled tube, and could not displace any water. The subject should instead choose the four small objects which would raise the water level sufficiently to bring the token within reach.

**Condition 3: Floating vs. Sinking objects.** Here, two identical medium sized tubes were presented, as used in condition 1. One set of heavy, sinking objects (clay spheres, 20 mm diameter), and one set of light, floating objects were presented (polystyrene spheres, 20 mm diameter). Heavy objects would sink and displace the water in the tube, whereas light objects floated on the surface of the water and were therefore non-functional. To make the objects visually distinct, one set was painted white and one painted black, with the colours counterbalanced across children. Unlike the other conditions, the relevant property here—weight—was not directly detectable through observation; therefore, in this condition the child was given the opportunity to handle each set of objects and place them into the object-tubes at the start of the trial.

**Condition 4: Hollow vs. Solid objects.** Here, two identical medium sized tubes were presented, as used in condition 1. One tube was presented with three grey hollow objects (metal cubes: 20 mm$^3$), and one tube with three grey solid objects of the same size and shape (clay cubes: 20 mm$^3$). Hollow objects consisted of a wire frame without solid sides (see Fig. 1) and therefore displaced only a small amount of water in the tube, but solid objects would raise the water sufficiently to bring the token within reach.

**Condition 5: Wide vs. Narrow tubes.** In this condition, the properties of the tubes were varied. One wide (diameter = 7 cm, height = 15 cm) and one narrow (diameter = 3.5 cm, height = 15 cm) water-filled tube were presented. The water level was equal for both tubes, and each were presented with an identical set of three medium sized grey objects in place (clay spheres: 20 mm diameter). If the subject chose to release objects into the narrow tube, the water level would rise by enough to bring the floating token within reach, but not in the wide tube.

**Condition 6: High vs. Low water levels in wide and narrow tubes.** This condition was identical to condition 5 except that the water levels in each tube varied. Here, the wide tube was presented with a higher initial water level than in condition 5, meaning that, now, objects released into the wide tube would bring the floating token within reach. The narrow tube was presented with a very low water level, and was therefore non-functional.

## Test procedure

On each experimental trial, two water-filled tubes were presented, each containing a floating out-of-reach token. One side of the tube was transparent, and one side was opaque. Tubes were initially presented with the transparent side facing the child, with the water level and token both visible. The object-tubes were pre-attached, and the subject observed as the experimenter inserted the objects into the object-tube (with the exception of condition 3: floating vs. sinking objects, where the children handled the objects and inserted them by themselves). In each condition, the water level was set so that for the correct choice, dropping the objects into the tube would raise the level sufficiently to allow the token to be removed, but not for the incorrect choice. The subject was asked to make their choice by pointing at the specific tube. The experimenter removed the tube that was not chosen. They then rotated the chosen water-filled tube so that the opaque side faced the subject, and indicated that the subject could now slide the barrier to release the objects into the water-filled tube. Hence, visual access was blocked after the choice was made, but before the resulting action of dropping the objects into the tube. The opaque side of the tube concealed the water level, but not the bottom of the tube (Fig. 1) to ensure that the child could see that the objects had rolled into the tube, but the objects' effects on the water level remained obscured. The experimenter then removed the object-tube, and the subject was able to attempt to fish the token from the tube of choice using the fishing rod.

Trials of each condition were presented in a pseudorandomised order across children. Each block of six trials contained one trial from each condition, with the provisions that the correct choice in any one trial (e.g., small objects) was not the same as in the previous trial, and that the correct side (left or right) was counterbalanced, ensuring that the correct choice was on the left three times and the right three times within each block of six trials, though not on the same side more than twice in a row within a block. The experimenter was RM or EL, with RM, EL or SAJ assisting during testing by re-setting tubes (emptying out the water and objects, replacing the water at correct level ready for next trial) as required. The experimenter followed a set script and procedure with each subject.

Conditions 1, 2 and 5, 6 were selected so that for each variation in size of tube or object, each option is correct on some trials but not on others, depending on the context of the

trial. For example, the narrow tube would be correct vs. the wide one when the water level was equal, whereas the wide tube would be correct vs. the narrow one when the water level was unequal.

## Data analysis

We recorded the choice per trial for each subject as 'correct' or 'incorrect'. All test sessions were coded live as well as being video-recorded unless parental consent requested otherwise. 10% of trials were coded from video and compared to the live coding, finding 100% agreement with the data. The full data set is available on Figshare: https://doi.org/10.6084/m9.figshare.3899787.v1.

We conducted Generalized Linear Mixed Models (GLMM: *Baayen, 2008*) using R (version 2.15.0; *R Core Team, 2014*) to assess which factors influenced success rate in the children. Success was a binary variable indicating whether the subject correctly solved the trial (1) or not (0), and was entered as a dependent variable in the models. We ran two models as we had two measures of interest: model (1) success on the first trial and model (2) success across all five trials. We included the random effect of subject ID, fixed effects of age in years (continuous: ages 5–9 in individual years), condition (1–6), gender (male/female), trial number (1–5; model 2 only) and the interaction between age and condition, and age and trial number (model 2 only). We used likelihood ratio tests to compare the full model (all predictor variables, random effects and control variables) firstly with a null model, and then with reduced models to test each of the effects of interest (*Forstmeier & Schielzeth, 2011*). The null model consisted of random effects, control variables and no predictor variables. The reduced model comprised of all effects present in the full model, except the effect of interest (*Göckeritz, Schmidt & Tomasello, 2014*). We then ran further analyses using exact two-tailed Binomial tests to assess success rate in each condition across two age groups (5–7 years & 8–9 years). We selected these two age groups to allow for easy comparison to previous Aesop's Fable studies (e.g., *Cheke, Loissel & Clayton, 2012*; *Miller et al., 2016*), whilst also minimising the possibility of Type II errors due to small samples size. Further analyses assessing success rate across all subjects (Table S1), and per age in years (Table S2) are presented in the supplementary materials.

## Ethics statement

The study was conducted under the European Research Council Executive Agency Ethics Team (application: 339993-CAUSCOG-ERR) and University of Cambridge Psychology Research Ethics Committee (pre. 2013.109). Informed written consent was obtained from parents prior to participation of the child. The parents of the child identified in the supplementary movie gave their informed written consent for this information to be published.

## RESULTS

On trial 1, the full model differed significantly from the null model ($X^2 = 19.39$, $df = 3$, $p = 0.0002$). We found a significant main effect of age ($X^2 = 8.92$, $df = 1$, $p = 0.002$: Table 1) on success rate (correct vs. incorrect choice), with success on the first trial increasing with

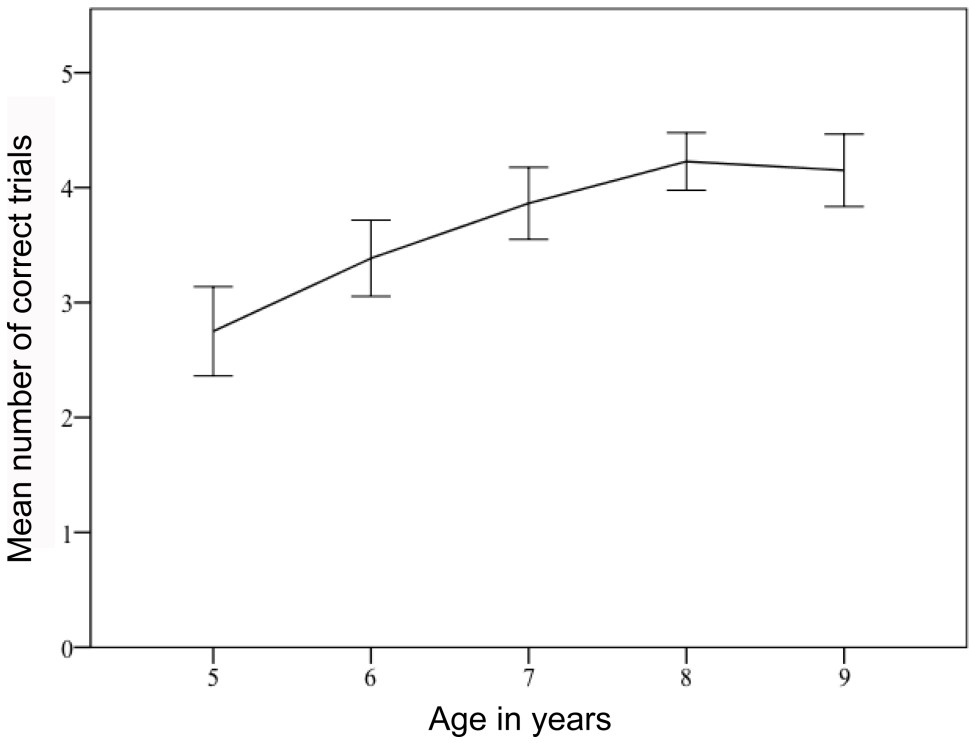

**Figure 2** **Mean number of correct trials across all 5 trials for all conditions by age (in years).** Success rate increased significantly with age. Error bars indicate standard errors.

**Table 1** **Generalized linear mixed models on factors affecting the number of correct trials in trial 1 (model 1) and across all trials (model 2).**

| Fixed term | Full model 1: trial 1 | | | Full model 2: all trials | | |
|---|---|---|---|---|---|---|
| | Estimate | z-value | p-value | Estimate | z-value | p-value |
| Age in years | **0.6072** | **2.876** | **0.004** | **0.5995** | **4.008** | **<0.001** |
| Condition | 0.435 | 1.192 | 0.233 | **0.4754** | **2.673** | **0.008** |
| Age*Condition | −0.0581 | −1.1 | 0.271 | **−0.0767** | **−2.909** | **0.004** |
| Gender | 0.4348 | 1.129 | 0.259 | 0.0163 | 0.082 | 0.934 |
| Trial number | / | / | / | 0.0085 | 0.040 | 0.968 |
| Age*Trial number | / | / | / | 0.0377 | 1.167 | 0.243 |

age. Across all five trials, we also found that the full model differed significantly from the null model ($X^2 = 74.62$, $df = 5$, $p < 0.001$). We found significant main effects of age ($X^2 = 16.42$, $df = 1$, $p < 0.001$: Table 1) and condition ($X^2 = 7.17$, $df = 1$, $p = 0.007$), and a significant interaction effect of age and condition ($X^2 = 8.5$, $df = 1$, $p = 0.004$) on success rate (correct vs. incorrect choices per trial). Success rate increased with age (Fig. 2), and success across all five trials was significantly poorer in the wide vs. narrow tubes condition, compared with the other conditions. Considering all subjects together, and over all five trials, children chose the correct option significantly more often that chance in all conditions except narrow vs. wide tubes (Table S1).

**Table 2** Correct choices (%) in each condition by each age group: 5–7 years old ($n = 34$) and 8–9 years old ($n = 21$). P-values ('$p$') are calculated from exact two-tailed binomial tests. Significant $p$-values are highlighted in bold.

| Age group | Large vs. small | | Too large vs. small | | Floating vs. sinking | | Hollow vs. solid | | Wide vs. narrow | | High vs. low | |
|---|---|---|---|---|---|---|---|---|---|---|---|---|
| | % | $p$ | % | $p$ | % | $p$ | % | $p$ | % | $p$ | % | $p$ |
| **Trial 1** | | | | | | | | | | | | |
| 5–7 | 76 | **0.003** | 29 | 0.024 NS | 56 | 0.608 | 53 | 0.864 | 38 | 0.229 | 79 | **0.001** |
| 8–9 | 95 | **<0.001** | 48 | >0.999 | 90 | **<0.001** | 86 | **0.002** | 62 | 0.383 | 81 | **0.007** |
| **Across all trials** | | | | | | | | | | | | |
| 5–7 | 62 | **0.002** | 71 | **<0.001** | 75 | **<0.001** | 69 | **<0.001** | 41 | 0.017 NS | 85 | **<0.001** |
| 8–9 | 91 | **<0.001** | 80 | **<0.001** | 94 | **<0.001** | 86 | **<0.001** | 65 | **0.003** | 87 | **<0.001** |

Notes.
NS, not significant with a Bonferroni correction.

We further explored correct choices within each condition per age group (5–7, 8–9 years old; Table 2). In condition 1 (large vs. small objects) and condition 6 (high vs low water level), children made significantly more correct than incorrect choices on *trial one* in both the 5–7 and 8–9 years old age groups. In condition 2 (too large vs. small condition), children significantly made the correct choices *across all trials* at both age 5–7 years and 8–9 years, but not on trial 1. In condition 5 (wide vs. narrow tubes), only children aged 8–9 years significantly made correct choices *across all trials*, and not from trial 1.

For the two conditions that have been previously tested in children using the standard Aesop's Fable task: floating vs. sinking and hollow vs. solid objects, we found that children made significantly more correct choices *across all five trials* in the age 5–7 year group, and from *trial one* in the 8–9 year group (Table 2). These results are similar to previous findings testing the sinking vs. floating condition in a standard Aesop's Fable paradigm (*Cheke, Loissel & Clayton, 2012*; *Miller et al., 2016*), but the children performed better than in previous tests using solid vs. hollow objects (*Miller et al., 2016*), which found that children struggled to select the correct option over five trials, but learnt to do so over twenty trials.

## DISCUSSION

In the current study, we developed a forced-choice Aesop's Fable paradigm, comprised of six different conditions, which could not be solved by responding only to perceptual-motor feedback. Subjects were not able to observe the water level rising when objects were dropped into the water-filled tubes, and had to select only one set of objects or one type of tube, into which all objects were dropped at once. To solve these tasks on their very first trial, children likely needed to mentally simulate the effect that dropping objects would have on the water level of the tubes, or, at a minimum, children needed to use prior semantic knowledge of the relevant properties of objects or tubes, to choose correctly on their first trial. Over five trials, here, children received feedback about the success of their actions at the end of each trial (whether the token could now be reached using the fishing rod), but at no point did they observe the water level rising. This contrasts with previous studies in which subjects were able to observe the change in the water level, within a trial, each time an object was dropped into a tube.
We found that there was a significant effect of age on success (correct vs. incorrect choices per trial) in trial one and across all trials, with success rate increasing significantly with age. We also found a significant effect of condition on success across all trials, and an interaction effect of age and condition across trials. Notably, the children's performance in the current paradigm followed a similar developmental pattern to that found on previous versions of Aesop's Fable tasks, where responding to perceptual-motor feedback had been a possible strategy for success (*Cheke, Loissel & Clayton, 2012*; *Miller et al., 2016*). We found that children aged 8–9 years passed the majority of conditions on their first trial (four of the six conditions), although 5–7 year olds passed two conditions on their first trial, and learnt to solve three of the four remaining conditions over five trials. This is comparable to standard Aesop's Fable tasks, which eight year old children typically pass on their first trial and 5–7 year olds can learn to pass over five trials. The finding that children's performance was not impaired in the current study, relative to previous standard Aesop's Fable tasks, indicates that children do not require visual feedback of the water level rising to solve these types of water displacement tasks.

Our results also highlight that the six conditions we presented were not equally easy for children to solve, and therefore may have each tapped slightly different cognitive processes. Younger children, aged 5–7 years, were able to pass the large vs. small condition and the high- vs. low-water level conditions on their first trial. These two conditions have not previously been used with children, though corvids have also consistently solved versions of these tasks over a small number of trials (*Bird & Emery, 2009*; *Taylor et al., 2011*; *Jelbert et al., 2014*; *Logan et al., 2014*). Given that younger children passed these two conditions only, it is possible they may have been solved using simpler mechanisms than the other variations of the task. For example, young children may have selected the tube with a high-water level simply because the token was already closest to the top of this tube, not because they imagined the effect that dropping objects would have on the future water level. Equally, young children may have had a general preference for the larger objects. There is some support for this as 5–7 year old children also selected the large objects more often than the small objects on the too large vs. small condition, where the large object could not fit into the narrow tubes and was therefore non-functional (though this trend was non-significant with a Bonferroni correction). Based on this pattern of performance, the evidence that children aged 5–7 mentally simulated the effects of dropping objects into the tubes is equivocal.

Older children, aged 8–9 years, were able to solve both these tasks, and additionally were able to pass the floating vs. sinking objects condition and the hollow vs. solid object condition on their first trials. Younger children learnt to solve these two conditions within five trials. Here, the developmental pattern on the floating vs. sinking condition is entirely in line with previous research (*Cheke, Loissel & Clayton, 2012*), while children's performance on the hollow vs. solid objects task was actually better than that observed in a previous study (*Miller et al., 2016*). In a standard version of the Aesop's Fable task, in which children chose to insert solid and hollow objects into a water-filled tube one at a time, Miller and colleagues found that 5–7 year old children learnt to select solid over hollow objects over the course of 20 trials, and though 8–10 year olds solved the task within five trials, they did not

do so from their very first trial. One explanation for the children's superior performance on the present task is that children in the current study were allowed to select one type of object only, which may have simplified their decision-making process. In line with this, in the earlier explorative task making some mistakes would not typically prevent the subject from obtaining the reward; thus, there was no penalty for testing out both of the presented options in early trials. Another possibility, which cannot be ruled out here, is that children's performances could have been scaffolded by their experience in the other concurrent test conditions. For example, when obtaining rewards using large or small clay spheres, children may have gained information that influenced their choices of solid over hollow cubes in this particular task. The information used could be simple, such as generalising the appearance of successful objects, or more complex, such as drawing inferences about the mechanics of water displacement from observing successes and failures in other contexts. While this is unlikely to account for first trial successes, given that trial orders were randomised, the opportunity to learn from other conditions over multiple trials may have contributed to children's ability to quickly acquire the correct option in the solid vs hollow task.

Performance in the remaining two conditions—wide vs. narrow tubes and too large vs. small objects—was significantly lower than in the other conditions, across all ages. In the too large vs. small condition children were less accurate on their first trial, but not over five trials. As discussed, there was a non-significant trend for 5–7 year old children to prefer the large object on their first trial, while 8–9 year old children chose at chance. This suggests that the children initially failed to recognise that the large object would not be able to fit into the narrow tube. However, once they had experienced this surprising event, they rapidly learnt to avoid choosing large objects on subsequent trials of this condition. Given that the difference in the sizes of the narrow tube and the large objects was quite subtle, it may be the case that children would solve this condition from the first trial if the disparity had been greater. In the narrow vs. wide condition performance remained significantly poorer than in other conditions both on the first trial of the task and over five trials. This condition has not been previously used with children; however, when tested with corvids, the majority have also struggled on this task (*Jelbert et al., 2014*), though some passed when the number of objects was restricted (*Logan et al., 2014*). The poor performance observed here strongly suggests that children between five and nine years are not able to accurately simulate the different effects that objects will have on the water level of differently sized tubes in this context. Thus, although by the age of seven, children can recognise that water volume is conserved when it passes between two containers of different sizes (*Piaget, 1930*; *Piaget, 1974*), they do not appear to possess a full, intuitive understanding of the behaviour of liquids in different containers.

Overall, the results reported here suggest that young children's success on Aesop's Fable tasks cannot be attributed to learning from perceptual-motor feedback because they did not observe the reward moving incrementally closer after each object drop. Furthermore, there was no evidence that the children found this task more difficult than other versions of water displacement tasks; indeed, performance on this task, which restricted access to visual feedback, was equivalent to (or better than) performance on previous versions of the Aesop's Fable task. In evaluating these findings it is important to note that children did

receive feedback on the overall success of their choices at the end of each trial. They may also have gained some information from other visible features, such as estimating the final water level from peering into the tube (though this was difficult to judge from above) or observing the light objects floating, and the too-large objects becoming stuck in the tube. However, they did not observe the reward moving incrementally closer after each object drop. Therefore the perceptual-motor feedback explanation cannot account for children's success on Aesop's Fable tasks (*Taylor & Gray, 2009*; *Cheke, Bird & Clayton, 2011*; *Jelbert, Taylor & Gray, 2015*). The pattern of results we found, in which the majority of tasks were solved on the first trial by children over the age of eight, is also consistent with a number of studies suggesting that children reliably solve various innovative tool-use problems only at around 7–8 years of age (*Beck et al., 2011*; *Beck et al., 2016*; *Hanus et al., 2011*; *Nielsen, 2013*). This adds to the growing body of evidence that spontaneously recognising the causal relations involved in tool use tasks can be remarkably difficult for young children.

The perceptual-motor feedback hypothesis was first suggested as an alternative explanation for the impressive performance of corvids that appear to demonstrate causal reasoning on the Aesop's Fable tasks (*Taylor & Gray, 2009*; *Cheke, Bird & Clayton, 2011*; *Jelbert, Taylor & Gray, 2015*). To date, the Aesop's Fable task has been used to assess the cognitive abilities of various species of corvid, including rooks. New Caledonian crows, Eurasian jays and California scrub jays (*Bird & Emery, 2009*; *Cheke, Bird & Clayton, 2011*; *Taylor et al., 2011*; *Jelbert et al., 2014*; *Logan et al., 2014*; *Logan et al., 2016*; *Miller et al., 2016*), as well as grackles, another behaviourally flexible species of bird (*Logan, 2015*; *Logan, 2016*). A number of great apes have also been tested on the comparable floating-peanut task (*Mendes, Hanus & Call, 2007*; *Hanus et al., 2011*). Recently, Miller and colleagues demonstrated that in New Caledonian crows, but not in human children, performance on object-choice tasks can be influenced by pre-existing preferences for certain types of objects, casting some doubt on the suggestion that birds' success on Aesop's Fable tasks reflects causal understanding—at least when considering their selection of objects (*Miller et al., 2016*). Perceptual-motor feedback has been suggested to underpin spontaneous string pulling behaviour performed by birds (*Taylor et al., 2010*; *Taylor, Knaebe & Gray, 2012*; see also *Jacobs & Osvath, 2015*; *Hofmann, Cheke & Clayton, 2016*, as well as performance on certain problem-solving tasks by great apes *Völter & Call, 2012*). However, to date, it is unclear whether the opportunity to receive perceptual-motor feedback accounts for non-human animals' ability to rapidly solve various water displacement tasks. The methodology that we describe here could be adopted for use with non-human animals to test whether their success depends on perceptual-motor feedback, with the present study allowing for comparison with young children. Use of this paradigm would help us to understand the learning mechanisms that might underpin the remarkable performance of certain species on water displacement tasks.

## ACKNOWLEDGEMENTS

We would like to thank the staff, parents and children at Great Abington Primary School, Holywell C of E Primary School, Sutton C of E Primary School, Stretham Community

Primary School and Spinney Primary School in Cambridgeshire for their participation in this study. Thank you to Ian Millar for help in apparatus construction, and to Markus Boeckle for assistance with the statistical analysis. We are also very grateful to Jennifer Vonk (Editor), Eva Reindl and an anonymous reviewer for constructive comments on the manuscript.

### Funding

Rachael Miller, Sarah A. Jelbert, Elsa Loissel, and Nicola S. Clayton received funding from the European Research Council under the European Union's Seventh Framework Programme (FP7/2007-2013)/ERC Grant Agreement No. 3399933, awarded to NSC. Alex H. Taylor was supported by a Royal Society of New Zealand Rutherford Discovery Fellowship. The funders had no role in study design, data collection and analysis, decision to publish, or preparation of the manuscript.

### Grant Disclosures

The following grant information was disclosed by the authors:
European Union's Seventh Framework Programme: FP7/2007-2013.
ERC Grant Agreement: 3399933.
Royal Society of New Zealand Rutherford Discovery Fellowship.

### Competing Interests

The authors declare there are no competing interests.

### Author Contributions

- Rachael Miller conceived and designed the experiments, performed the experiments, analyzed the data, wrote the paper, prepared figures and/or tables, reviewed drafts of the paper.
- Sarah A. Jelbert conceived and designed the experiments, performed the experiments, wrote the paper, prepared figures and/or tables, reviewed drafts of the paper.
- Elsa Loissel conceived and designed the experiments, performed the experiments, reviewed drafts of the paper.
- Alex H. Taylor conceived and designed the experiments, reviewed drafts of the paper.
- Nicola S. Clayton conceived and designed the experiments, reviewed drafts of the paper.

### Human Ethics

The following information was supplied relating to ethical approvals (i.e., approving body and any reference numbers):

The study was conducted under the European Research Council Executive Agency Ethics Team (application: 339993-CAUSCOG-ERR) and University of Cambridge Psychology Research Ethics Committee (pre.2013.109). Informed written consent was obtained from parents prior to participation of the child.

## Data Availability

Miller, Rachael; Jelbert, Sarah; Loissel, Elsa; Clayton, Nicola; Taylor, Alex (2017): Miller, Jelbert et al. Young children do not require perceptual-motor feedback to solve Aesop's Fable tasks. figshare. https://doi.org/10.6084/m9.figshare.3899787.v1.

## Supplemental Information

Supplemental information for this article can be found online at http://dx.doi.org/10.7717/peerj.3484#supplemental-information.

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
