# Peer review of "Young children do not require perceptual-motor feedback to solve Aesop’s Fable tasks"

_PeerJ, doi:10.7717/peerj.3484_

## Round 0.1 · original submission · Minor Revisions

· Academic Editor

Minor Revisions

Apologies for the slightly extra time taken to find reviewers for your article. I was fortunate to eventually find two expert reviewers who had very favorable impressions of your manuscript, and who provided extremely thorough reviews. Please revise your manuscript according to the reviewers’ comments and return the revised MS along with a detailed cover letter indicating how you’ve addressed the comments. For example, both reviewers make several suggestions regarding the need to make various elements of the procedure more explicit for the reader. Please attempt to explain the rationale behind each design feature cIearly for readers less familiar with the paradigm. However, I think it is clear why the tube was opaque on one side and transparent on the other.

I have a few additional minor comments of my own:

I agree with the reviewers that the manuscript is well written and clear and I appreciate the goal of elucidating the mechanisms underlying performance compared to the more typical focus on task performance alone.

I understand why the children are lumped into categorical age groups that map on to the previous literature; however, treating age as a discrete rather than a continuous variable is a bit arbitrary and causes some loss of variance in the analysis. Ideally, age would be treated as a continuous variable in the analyses (by months). Such an approach might also make it more clear if there is a sudden or gradual shift in understanding over development. The paper treats age as if there are two categories (younger and older children) when the data as plotted in Fig. 2 makes more sense. In that figure, it is not clear that there is a clear distinction between two groups as described.

It isn’t clear why the stone option in the ‘sawdust vs. water condition’ wouldn’t be correct – why can’t the children use a magnet to retrieve an object from that tube? I also agree with the reviewer that it is odd to label this “the sawdust..” condition when it does not involve sawdust.

Please be clear on whether the order of conditions was randomly presented across children.

If all trials were videotaped you must report reliability analyses for the coded measure of choice.

It would be interesting to include an interaction term in the statistical models between age and trial number.

For the y axis label on Figure. 3, please relabel as “mean number of successful choices”. “Accuracy” implies a proportion or percentage correct.

Small Points:

I don’t really understand the “token trail”
Change “which” to “that” on line 20 in the abstract and on line 87.
Delete the “or not” on lines 22 and 34, 45 and elsewhere.
Indicate how many children were tested on line 23.
The reference to the relevant study should appear immediately at the end of line 41.
Change “than” to “that” on line 55
Insert a comma after ‘7’ on line 395
Move the “only” on line 404 to after ‘problems’ on line 405.
Why not reference the grackle studies (Logan, 2015;2016) in the Introduction as well indicating how they performed relative to the corvids?

·

Basic reporting

This is a well-written paper investigating young children’s causal understanding of water displacement using the Aesop’s fable task. Previous studies had shown that 5- to 7-year-olds are able to learn how to solve this task; however, it was unclear whether children’s success was based on a true causal understanding of water displacement or whether it could be explained by a simpler process such as learning through perceptual-motor feedback. The current study focused on this “perceptual-motor feedback hypothesis”, investigating whether young children could solve the Aesop’s fable task when access to perceptual-motor feedback was prevented.

This article is a unique contribution to the literature as it is the first to further investigate children’s causal understanding in the Aesop’s fable task, trying to rule out an important alternative hypothesis. In addition, the task was designed and seems suitable for an adaptation to non-human subjects, which is another valuable contribution to the field.
The manuscript is well written, the arguments are easy to follow, and the language is professional and clear. As I am not a native speaker I cannot comment on fine language details, but clearly there are no major languages mistakes. The title summarizes the study well.

However, I have a few comments with regard to slightly ambiguous or unclear labels and expressions:

1. In the abstract, it does not become clear how the removal of the visual feedback is actually achieved. The forced-choice paradigm is mentioned, but it remains unclear what it consists in and how it is relevant for the experimental manipulation. I suggest inserting a sentence explaining in how far the removal of visual-motor feedback is achieved by applying a forced-choice paradigm. In addition, if space allows, you could also state that the paradigm involved dropping all objects at once, which is another, related means for removing perceptual-motor feedback

2. Lines 79-81: I think the introduction would benefit if the authors could state more explicitly what is meant by “performance” and “success” on the trials. I guess both success on the first trial and success across trials (after learning) are the subjects for explanations (because both are investigated in the current study), but I think the authors should present the explanations of the “causal understanding hypothesis” and the “response to perceptual-motor feedback hypothesis” separately for success on the first trial and for learning performance. In previous studies, success on first trial could have been due to a real causal understanding and a simulation of the objects of the objects but also due to response to perceptual-motor feedback. However, in the current study – in which all objects are dropped at once – success on the first trial can by design only be explained via a causal understanding. This difference should be stated more explicitly, also because it explains how the design of the current study aims to test the “response to visual-motor feedback hypothesis”. Apart from success on the first trial, the learning rates are a separate issue. As clearly stated by the authors, the learning found in previous studies can be explained by both hypotheses. The current study also tests for learning across trials and in my understanding it is claimed that children would not be able to learn across trials via response to visual-motor feedback as they could only receive feedback about the success of their actions. As mentioned in my point below on the discussion section, the authors need to discuss (not necessarily in the introduction, but definitely in the discussion) in how far their study succeeds in preventing children from learning from perceptual-motor feedback across trials and in how far “learning from ones’ actions” is different from learning through perceptual-motor feedback (see also my comment below).

3. In the methods sections and all parts thereafter: Why did the authors choose to label their outcome variable “accuracy”? To me, this label is potentially confusing as accuracy might suggest that there is a degree of “how much one can get something right”. In the current paper however, the variable described is binary (trial correct: yes, no). Why not use the label “success” instead of accuracy? I would suggest naming the the first outcome variable “success on first trial”, the second one “success rate across trials”

4. Line 182: the label “sawdust” condition is confusing because actually no sawdust is used. I understand why the authors used it (to draw a connection to previously used trials), which I think is a good point. However, I would prefer if the authors put the term “sawdust” in quotation marks

Minor language issues:

1. Line 55: typo: “that” instead of “than”

2. Line 224: maybe better to write “children” instead of “child” because later in the sentence it is also written “themselves”, so plural might be more appropriate

3. Line 248: better write “was not” instead of “wasn’t”, as this is a bit informal

4. Lines 310/311: maybe rephrase “from age 5-7” and “from age 8-9” to either “in age group 5-7” or “from age 5 to 7”, but avoid mixing words (“from”) with symbols (“-“).

5. Lines 427 and 503: typo: “Völter” instead of “Volter”

Regarding the background literature, the information presented is detailed, complete, and gives a concise overview of the current literature. The authors place their current study well into context and thus make their paper also accessible for non-experts. The paper explains clearly the current open questions in the field as well as the resulting research question of the current study.

I have one remark with regard to the references presented in the third sentence of the introduction (lines 43/44, i.e., Mendes et al 2007; Hanus et al 2011): I would suggest moving these references to the end of the preceding (second) sentence (after: “in order to bring a floating peanut within reach”) because the second sentence mentions studies on both humans and great apes (so the references would be appropriate there) whereas the third sentence focuses only on the child study (Hanus et al 2011). Therefore, I would include Mendes et al 2007 and Hanus et al 2011 as references into the second sentence and in the third sentence only refer to Hanus et al 2011 (and thus delete the great apes study (Mendes et al. 2007)).

With regard to the figures, I compliment the authors on Fig. 1 which is very well designed and which is really helpful for understanding the procedure, the experimental conditions, and what the correct choice in each condition is. The caption for Fig. is written really well.

With regard to Fig. 2, I have the following comments:

1. In the caption of Fig. 2 it should be explained what the vertical bars stand for (standard errors?)

2. Related to my point above regarding the labelling of the outcome variables (accuracy vs success), I suggest labelling the y-axis in Fig. 2 and 3 something like “number of correct trials” or “number of successes”, which in my view is even more intuitive than accuracy

Minor issues with regard to the figures:

1. Since figures are usually meant to stand for themselves, I suggest changing the label of the x-axis in Fig. 2 from “age” to “age in years”

2. Since figures are usually meant to stand for themselves, the caption of Fig. 3 could include the information that condition 5 was significantly different from the other conditions

Thank you for sharing the raw data. The video file is enlightening and useful. If the authors have permission to do so, the authors could even provide a second example video or extend the first one.

The article represents an appropriate unit of publication.

Experimental design

The article describes original primary research. The research question is well defined and the authors explain how their paper fills the identified knowledge gap.

However, with regard to the research rationale, I suggest that the authors explain more explicitly how their study attempts to prevent participants to learn through visual feedback. This information could be added to the end of the introduction (somewhere between lines 115 and 142). It seems to me that the experimental manipulation consists of several parts which all contribute to removing opportunities for learning from visual-motor feedback. First, there is the forced-choice paradigm which aims to encourage children to simulate the effect of the objects in advance. Second, the objects are all released at once. Lastly, the tubes are occluded and are turned round after children indicated their choice. In their discussion, from lines 320 to 326, the authors give an excellent description of the study rationale and I would suggest to move this paragraph to, or replicate it, in the introduction to support the reader’s understanding of the study rationale.

The methods section is well described and also explains relevant details that might be obvious to the authors but less clear to the reader, for example the explanation of why a fishing rod was used (to ensure a standardized distance to the water, lines 163/164). However, more information needs to be presented in order to enable researchers to replicate the experiment:

1. If possible, please provide information on the ethnic composition of the sample, and on the socioeconomic status. This is important for potential meta-analyses and because the role of context in human development becomes more evidenced

2. State whether there were any dropouts and if yes, why.

3. Why was this sample size chosen? Was an a priori sample size calculation performed?

4. In the second step of the pre-training, which objects were used as training objects?

5. Also related to the second step of the pretraining, I assume that the tubes were also transparent on one side and opaque on the other and that they were turned after children made a choice? Please add this information in a sentence, it might not be obvious to readers.

6. In the methods section or maybe at the end of the introduction, please provide some more information on why you chose a version of the task in which the objects would be dropped all at once rather than a version which is closer to the previous studies – in that children drop the objects one by one but where visual feedback from the tube is still occluded (by having one side of the tube opaque).

The experiment was conducted to a high technical and ethical standard. Informed written consent was obtained from the participants’ parents.

With regard to data analysis, I have several comments:

1. Line 277: It is not explained how “accuracy across all five trials” was calculated (I assume this is the sum of successes across all 5 trials) and this information needs to be added as this might not be obvious.

2. Line 277: It remains unclear whether age was entered into the GLMM as a categorical or a continuous variable. Please add the missing information.

3. Line 279: Information should be added with regard to which post-hoc tests were carried out to determine which conditions were significantly different from each other

4. Why was a stepwise selection method chosen? To me it seems that your primary research question was one about hypothesis testing (whether condition and age have a significant effect on children’s success) rather than about an exploratory approach to finding the “best” model. More importantly, however, I would like to draw the authors’ attention to a publication by Mundry and Nunn (2009; and references herein) amplifying “previous warnings about using stepwise procedures” (p. 119) concluding that these procedures “should not be used in the context of testing null hypotheses about a set of predictor variables” (p. 121). This is because stepwise methods lead to greatly inflated type 1 error rates due to multiple testing. I therefore suggest removing the stepwise selection procedure from the data analysis.

Reference: Mundry, R., & Nunn, C. L. (2009). Stepwise Model Fitting and Statistical Inference: Turning Noise into Signal Pollution. The American Naturalist, 173(1), 119-123.

5. Studies using GLMMs usually first conduct a test of a full model against a null model (i.e., a model containing all control variables and random effects but lacking the crucial predictor variables (which in the current study would be age and condition (and trial number)) and only if this analysis reveals a significant result, effects of individual predictors should be analysed (see Forstmeier & Schielzeth, 2011; for examples of such approaches see Göckeritz, Schmidt, & Tomasello, 2014; Kalan, Mundry, & Boesch, 2015; Völter, Sentís, & Call, 2016). I therefore suggest running and reporting a test of the full model before presenting the analysis for the individual predictors.

References:
Forstmeier, W., & Schielzeth, H. (2011), Cryptic multiple hypotheses testing in linear models: overestimated effect sizes and the winner’s curve. Behavioral Ecology & Sociobiology, 65, 47-55.
Göckeritz, S., Schmidt, M. F. H., & Tomasello, M. (2014). Young children’s creation and transmission of social norms. Cognitive Development 30, 81–95.
Kalan, A. K., Mundry, R., & Boesch, C. (2015). Wild chimpanzees modify food call structure with respect to tree size for a particular fruit species. Animal Behaviour 101, 1-9.
Völter, C. J., Sentís, I., & Call, J. (2016). Great apes and children infer causal relations from patterns of variation and covariation. Cognition 155, 30–43.

With regard to the results section, I have the following comments:

1. Line 295: In the second sentence of the results section (lines 293-295), it is stated that conditions 2 and 5 differ significantly from the other conditions, and the reader gets referred to Table 1. However, Table 1 does not list the results of the comparisons of the conditions against each other, only the estimate for condition overall. Therefore, I recommend that these missing details (estimates on the individual conditions compared against each other) should be added to Table 1.

2. Line 299: Similar to point 1, where the text explains that condition 5 is significantly different from the other conditions, the reader is referred to Table 3 – again, this information cannot be found in the table and needs to be added

3. Lines 301/302: More information should be provided with regard to how the analysis of correct choices within each condition per age group was performed. So far, the information that binomial tests were used can only be found in the caption of Table 3. I would recommend to provide this information in the results section (possibly also in the data analysis section) as well.

4. Related to point 3, it would be good if the authors could explain why they chose to use binomial tests for this analysis. I wonder whether the same or a similar analysis could have been included into the GLMM (maybe via an interaction) and maybe this would have avoided multiple testing and the need for a Bonferroni correction. However, maybe the sample size would have been too small for including an interaction term into the GLMM. It would be great if the authors could provide more information on why they decided to use this analysis (I do not mean to imply that it is wrong).

Validity of the findings

The conclusion drawn in the current study is clearly stated (lines 335-337), in that children do not require perceptual-motor feedback to solve the Aesop’s fable task. In addition, the study also replicates the developmental pattern found in previous studies (older children pass the task on their first trials, younger ones do not do so). Therefore, the study is also a valuable contribution to the literature with regard to replicating earlier findings.

The discussion section is well-written, it balances strengths and limitations of the study, discusses all major findings, and suggests possible explanations for them. Specifically, I compliment the authors on their discussion of the six conditions not being equally easy and what this could mean with regard to the cognitive mechanisms involved (lines 339-355). The authors also link their findings to the existing literature and suggest possible explanations for the differing results (lines 386-388).

However, I have a few comments on the discussion section:

1. I would like the authors to elaborate on the potential criticism that participants in their study were still able to solve learn the task via visual-motor feedback. One could argue that even though children did not see the water level rising, they do see the “new” water level when they use the fishing rod (at least the girl in the example video provided is looking down the tube). It seems that by being able to see the water level after the objects have been dropped, children are still able to receive visual feedback from the task. Even though they do not actually see the water level rising, they might be able to mentally “reverse-engineer” what was happening when they see the new water level. Another point could be that when children see that the stones that are too large get stuck in the tube this could also provide some kind of perceptual-motor feedback for the children. I think the discussion section would benefit from including this point: In how far might children still have been able to learn via perceptual-motor feedback?

2. Somewhat related to point 1: In lines 324-326, the authors state “Over five trials, children received feedback about the success of their actions (whether or not the token could now be reached using the fishing rod), but at no point did they observe the water level rising.” I would like the authors to elaborate somewhat on this statement. What does it mean that children were able to learn from their actions? Could that be seen as a form of learning via perceptual-motor feedback as well? For example, as indicated above, when children see the objects that are too large getting stuck in the tube, is that not an example of perceptual-motor feedback? Where exactly is the difference between learning from the success of one’s actions and learning from perceptual-motor feedback? I find this especially important to answer because in the current study even if the tubes would not have been opaque, due to the fact that all objects were dropped at once learning from one’s actions (successfully solved task: yes or no) and learning from perceptual feedback from the rising water level (water level risen: yes or no) coincide (whereas in previous studies because objects had to be dropped repeatedly, feedback from the rising water level and feedback about the success of one’s actions was somewhat independent).

3. Lines 399-400: Given the above point with regard to whether it could still be possible that children were able to learn how to solve these tasks via perceptual-motor feedback, the sentence in lines 399-400 might have to be rephrased slightly. I would have written “Overall, the results reported here suggest that young children’s success on the first trial of Aesop’s Fable tasks cannot be attributed to learning from perceptual-motor feedback.” I think this is a conclusion that is clearly supported by the current findings, but I am less convinced that the current study ruled out the possibility that children’s learning took place without visual-motor feedback of some sort.

4. Line 405: It is great that the authors draw a connection to literature from a different field (innovative abilities in children) but I think another sentence would be needed to explain why this literature was mentioned. What do these studies and the current study have in common? Do they hint at the same cognitive processes involved?

Minor points in the discussion:

1. Regarding the discussion of the “narrow vs wide tube” condition (lines 388-398): When reading through the methods and looking at Fig. 1, I found the solution in the “narrow vs wide tube” condition not as intuitive as the solutions in the other tasks. I think it might be premature to conclude that children’s poor performance in this task indicates a lack of understanding (“are not yet able to accurately simulate the different effects that objects will have on the water level of differently sized tubes”, lines 393-395). It might be worthwhile to consider testing this condition with a small sample of adults to see how they perform on this task (not necessarily for this revision, but it should be considered for the future). If children’s poor performance is actually due to a lack of understanding, one would expect adults to show excellent performance on first trial, but if adults struggle as well, one might have to question the validity of the task and to adjust it accordingly – the authors themselves already acknowledged that the difference between the two tubes was only subtle (line 387).

2. Line 365: It could be stated a bit more clearly what the procedure in Miller et al 2016 was. Specifically, it could be stated explicitly that children were allowed to select several types of objects – in the current manuscript, this information has to be inferred from line 365. So maybe state in half a sentence that children could insert both hollow and solid objects into the tubes.

Additional comments

This is a well-written manuscript with interesting findings that definitely deserves publishing. However, I suggest elaborating on some points in the introduction and discussion (see above) to enhance understanding for the reader, to add some more information in the methods and results section, and to consider removing the stepwise procedure.

Reviewer 2 ·

Basic reporting

This MS is generally very clearly and unambigously written. The Introduction is concise, and provides the context for the study, with the relevant literature clearly referenced. The structure appears to conform to PeerJ standards, and an accessible raw data set is supplied, along with a supplementary video of examples of two trials, which helps the reader to understand the forced-choice procedure.

All the figures and tables are relevant. Figure 1 is excellent, and makes it much easier for the reader to understand and compare the conditions used. However, Figures 2, 3 and Table 3 all need a small amount of additional information. Figures 2 and 3 need information in the legend to explain what the error bars are (I assume they are standard error bars, but it is not clear). Table 3 should state what the adjusted p-value threshold is following Bonferroni correction. Column headings are also inconsistently labelled as both upper and lowercase 'p'.

Where citations that cover a range of different species are mentioned, it would be helpful to identify which paper tested which species (e.g. the list given lines 71-72), as the authors do elsewhere (e.g. lines 90-93).

Experimental design

The research itself is well within the scope of the remit of PeerJ. The specific knowledge gap addressed (whether young children succeed in Aesop's fable tasks through perceptual-motor feedback) is clearly identified, and the elegantly designed set of experiments are appropriate to answer this question.

The experimental design is of a high standard, with a good sample size. The methods are mostly explained clearly and in sufficient detail, however there were a few points that remained obscure that I think the authors need to clarify.

1. Each of the tubes includes a clear section at the bottom, even when the opaque part of the tube is rotated. This is evident from Figure 1, and from the supplementary video. Since the top part of the tube is covered, children cannot see the water level rising (thus preventing access the main source of perceptual-motor feedback), but they can see (for example) the sinking objects appearing at the bottom, in contrast to floating objects. Can the authors explain the rationale behind leaving a portion of the tube uncovered at the bottom?
2. A first reading of lines 195-197 makes it appear that at least one of the conditions might be split across days, because 18 trials are completed in one day, and there are (apparently) 5 trials per condition. It is made clear that this is not the case in line 254, because - as far as I understand it - each 'block' is composed of 1 trial of each of the 6 conditions, intermixed. Thus an 18/12 split of trials allows for complete sets of these 6 trial blocks on each day. It would be helpful to explain the trial structure earlier to avoid this confusion. Also, more detail on the pseudorandom scheme used would be helpful.
3. It is not immediately clear whether the functional distinction between the hollow and solid cubes in Condition 4 is obvious from observing the objects. Looking closely at the cubes depicted in Figure 1D, it appears that the 'hollow' object is actually more like a wireframe cube, without solid sides, rather than the solid-sided but hollow interior object I had imagined. Altering the description in the text would avoid this confusion. line 223: typo - "clay cubes"
4. Are the ages given in line 147 school year ages or actual ages? It would be helpful to state the actual mean ages (plus range) in years and months, since there can be substantial variation within a school year group.
5. Line 281 ("Each dropped variable was then re-added separately to the final model to check it remained significant") is confusing. Perhaps the authors could elaborate slightly on the model selection process.

Validity of the findings

The analysis appears to be statistically appropriate, the results are well controlled, and the conclusions supported by the results, addressing the research question specifically.

However, I think that the authors should provide a rationale for grouping children into two age categories (5-7 and 8-9) for the analysis of choices within each condition (lines 301-315, Table 3), and specifically why this particular grouping was made rather than analysing each age separately.

Finally, the authors use the phrase "mentally simulate" several times to describe the putative cognitive mechanism that children (or other species) solving the task on the first trial without perceptual-motor feedback might be using (e.g. in the abstract, lines 82, 323, 354). I suspect that this is the most likely explanation, but since other mechanisms are possible, and the authors did not ask children to describe how they solved the task, I think this term is too specific. If used, it should be acknowledged that there are other potential mechanisms that would fit the observed results.

Additional comments

My only other comments not covered by the sections above are minor typos:

line 55: typo - "objects that sank"

line 273 "binary variable indicating whether"

line 303: "signficantly more correct choices on trial 1..." more than what?

---

## Round 0.2 · Minor Revisions

· Academic Editor

Minor Revisions

First, let me apologize for not getting to your paper immediately upon submission. I was travelling out of the country last week. I’ve now had a chance to read through your revision carefully. This is a well written paper and a well-designed experiment. Thank you for being responsive to the reviews during the previous round. I think the methods are much more clear with this revision and the logic follows very nicely. I have some very minor changes to suggest before I can formally accept the MS:

Lines 31, 41, 132, 464, 467, “or not” is not needed after “whether”.
Please write out numbers less than 10 in full (e.g., line 51) unless referring to a measurement.

In the PDF file, there is a missing empty line between paragraphs. Please ensure spacing is consistent in the final MS.

Please insert commas after i.e. and e.g. (e.g., lines 86, 271).

Heinrich and Bugnyar need not be italicized on line 91.

On line 175, should “trial” be “trail?”. You haven’t described the trail at this point yet so it’s a bit confusing. It’s referred to as a “token track” on line 188l. Please be consistent and begin by describing it in Materials before Procedure.

Can you rephrase on lines 278-279 – it currently reads as if there are trials in which both options are correct, which I don’t believe to be the case. Maybe “each option is correct on some trials but not others”.

On line 405, change “while” to “Although”

---

## Author Rebuttal · Round 0.2

Department of Psychology
University of Cambridge
Cambridge, UK
Phone: 01223 747321

14/05/2017

Dear Dr Vonk,

Thank you for the opportunity to make minor revisions to our manuscript *"Young children do not require perceptual-motor feedback to solve Aesop's Fable tasks".* We appreciate the detailed comments made by the two reviewers, and are happy to make the requested changes. We have responded point-by-point below, with our comments in blue. We have also made all of the requested small changes and language corrections suggested by the editor and both reviewers. **NB: Line numbers refer to the track-changes copy of the document.**

We hope that in this revision we have addressed all of the points raised by the reviewers and the editor and that the manuscript is now suitable for publication in PeerJ.

Yours sincerely,

Sarah Jelbert, Rachael Miller, Elsa Loissel, Alex Taylor and Nicky Clayton.

**Response to reviewers**

**Editor's comments**

I understand why the children are lumped into categorical age groups that map on to the previous literature; however, treating age as a discrete rather than a continuous variable is a bit arbitrary and causes some loss of variance in the analysis. Ideally, age would be treated as a continuous variable in the analyses (by months). Such an approach might also make it more clear if there is a sudden or gradual shift in understanding over development. The paper treats age as if there are two categories (younger and older children) when the data as plotted in Fig. 2 makes more sense. In that figure, it is not clear that there is a clear distinction between two groups as described.

** The GLMM's were run using age in years (continuous – since the concept of age is continuous) rather than age groups (older and younger children) to allow for a more detailed initial test of age effects on success rate. The Binomial exact two-tailed tests were then run using the simplified age group categories (5-7, 8-9 years) to allow for easy comparison to previous studies and avoid further multiple testing. We have now clarified these distinctions in the analysis more clearly within the text, and in Tables 1 and 2 (**Lines 326, 335** in data analysis). We feel that in order to allow for easy comparison to the previous studies and to present a general overview across younger-older children, it is clearer to continue to refer primarily to the results of the age group analysis within the results and discussion. However, we agree that it may be of interest to additionally include further analysis using age in years (i.e. similar to the age groups analysis shown in Table 2), and have now done so within the supplementary materials (**Table S2**). We still feel that the use of years as an age measure rather than months is most appropriate, in order to allow for clear comparison to previous related research (e.g. Cheke et al., 2012; Miller et al., 2016).

It isn't clear why the stone option in the 'sawdust vs. water condition' wouldn't be correct – why can't the children use a magnet to retrieve an object from that tube? I also agree with the reviewer that it is odd to label this "the sawdust.." condition when it does not involve sawdust.

**We have re-named this condition to pebbles vs. water, and now explicitly state that at the start of the test the level of both the pebbles and the water was too low for the child to reach the token with the magnetic fishing rod (**lines 216** onwards).

Please be clear on whether the order of conditions was randomly presented across children.
**We now state this in **line 300**.

If all trials were videotaped you must report reliability analyses for the coded measure of choice.
** 10% of videos were coded and compared to the results recorded at the time of testing, finding 100% agreement with the data. We have added this to **line 318**.

It would be interesting to include an interaction term in the statistical models between age and trial number.
** Good suggestion, thank you. We added this interaction to GLMM model 2 and found it to be non-significant (**Table 1**).

For the y axis label on Figure. 3, please relabel as "mean number of successful choices". "Accuracy" implies a proportion or percentage correct.
**We have relabelled the figures as requested, and no longer refer to 'accuracy' in the text.

**Basic Reporting**

In the abstract, it does not become clear how the removal of the visual feedback is actually achieved. The forced-choice paradigm is mentioned, but it remains unclear what it consists in and how it is relevant for the experimental manipulation. I suggest inserting a sentence explaining in how far the removal of visual-motor feedback is achieved by applying a forced-choice paradigm. In addition, if space allows, you could also state that the paradigm involved dropping all objects at once, which is another, related means for removing perceptual-motor feedback

**We have added this information to the abstract as requested.

Lines 79-81: I think the introduction would benefit if the authors could state more explicitly what is meant by "performance" and "success" on the trials. I guess both success on the first trial and success across trials (after learning) are the subjects for explanations (because both are investigated in the current study), but I think the authors should present the explanations of the "causal understanding hypothesis" and the "response to perceptual-motor feedback hypothesis" separately for success on the first trial and for learning performance. In previous studies, success on first trial could have been due to a real causal understanding and a simulation of the objects of the objects but also due to response to perceptual-motor feedback. However, in the current study – in which all objects are dropped at once – success on the first trial can by design only be explained via a causal understanding. This difference should be stated more explicitly, also because it explains how the design of the current study aims to test the "response to visual-motor feedback hypothesis". Apart from success on the first trial, the learning rates are a separate issue. As clearly stated by the authors, the learning found in previous studies can be explained by both hypotheses. The current study also tests for learning across trials and in my understanding it is claimed that children would not be able to learn across trials via response to visual-motor feedback as they could only receive feedback about the success of their actions.

**As requested, we now discuss success on the first trial, and success over 5 trials separately in the introduction (in terms of how that relates to causal understanding and perceptual-motor feedback) (**lines 138** onwards)

In the methods sections and all parts thereafter: Why did the authors choose to label their outcome variable "accuracy"? To me, this label is potentially confusing as accuracy might suggest that there is a degree of "how much one can get something right". In the current paper however, the variable described is binary (trial correct: yes, no). Why not use the label "success" instead of accuracy? I would suggest naming the the first outcome variable "success on first trial", the second one "success rate across trials"

**We now refer to 'correct choices' or 'success' rather than accuracy, throughout the manuscript.

Line 182: the label "sawdust" condition is confusing because actually no sawdust is used. I understand why the authors used it (to draw a connection to previously used trials), which I think is a good point. However, I would prefer if the authors put the term "sawdust" in quotation marks

**We have re-named this condition to pebbles vs water.

**Figures**

In the caption of Fig. 2 it should be explained what the vertical bars stand for (standard errors?)

Related to my point above regarding the labelling of the outcome variables (accuracy vs success), I suggest labelling the y-axis in Fig. 2 and 3 something like "number of correct trials" or "number of successes", which in my view is even more intuitive than accuracy

Since figures are usually meant to stand for themselves, I suggest changing the label of the x-axis in Fig. 2 from "age" to "age in years"

Since figures are usually meant to stand for themselves, the caption of Fig. 3 could include the information that condition 5 was significantly different from the other conditions

**Due to requested changes in the analysis we no longer include Figure 3 in the manuscript, but we have made all of the reviewer's suggested changes to **Figure 2**, including stating the error bars, re-labelling the y-axis to 'mean number of correct trials' and the x-axis to 'age in years', and stating that the success rate increased significantly with age.

**Experimental design**

I suggest that the authors explain more explicitly how their study attempts to prevent participants to learn through visual feedback. This information could be added to the end of the introduction (somewhere between lines 115 and 142). It seems to me that the experimental manipulation consists of several parts which all contribute to removing opportunities for learning from visual-motor feedback. First, there is the forced-choice paradigm which aims to encourage children to simulate the effect of the objects in advance. Second, the objects are all released at once. Lastly, the tubes are occluded and are turned round after children indicated their choice. In their discussion, from lines 320 to 326, the authors give an excellent description of the study rationale and I would suggest to move this paragraph to, or replicate it, in the introduction to support the reader's understanding of the study rationale.

**We are now more explicit about how our study design eliminates perceptual-motor feedback in the introduction (**lines 134-136**), as requested.

If possible, please provide information on the ethnic composition of the sample, and on the socioeconomic status.

**We did not record information on the ethnicity of participants, but we can report that this study was conducted in five schools which all served predominantly white, middle-class areas (**line 180**).

State whether there were any dropouts and if yes, why.

**There were no dropouts. All children completed both testing sessions (**line 178**).

Why was this sample size chosen? Was an a priori sample size calculation performed?

**This sample size was chosen to ensure we had a minimum of 10 children per age group, and included a similar number of participants to a previous Aesop's Fable study with children (Cheke et al., 2011) (**lines 176-178**).

In the second step of the pre-training, which objects were used as training objects?

**The blue oblongs were used throughout pre-training (**line 219**).

Also related to the second step of the pretraining, I assume that the tubes were also transparent on one side and opaque on the other and that they were turned after children made a choice? Please add this information in a sentence, it might not be obvious to readers.

**Yes, we now state that the tubes are the same as those used in the main test in **line 218**.

In the methods section or maybe at the end of the introduction, please provide some more information on why you chose a version of the task in which the objects would be dropped all at once rather than a version which is closer to the previous studies – in that children drop the objects one by one but where visual feedback from the tube is still occluded (by having one side of the tube opaque).
**We have added this to **lines 134-137**

**Data Analysis**

With regard to data analysis, I have several comments:

1. Line 277: It is not explained how "accuracy across all five trials" was calculated (I assume this is the sum of successes across all 5 trials) and this information needs to be added as this might not be obvious.
**We now no longer use the term 'accuracy' in the manuscript.

2. Line 277: It remains unclear whether age was entered into the GLMM as a categorical or a continuous variable. Please add the missing information.
**This was continuous. We have clarified this in the data analysis section in **line 327**.

3. Line 279: Information should be added with regard to which post-hoc tests were carried out to determine which conditions were significantly different from each other
**This was Binomial exact two-tailed tests. We now state this in the data analysis section in **line 334-340,** and in the results **line 370-372** and **Table S1**.

4. Why was a stepwise selection method chosen? To me it seems that your primary research question was one about hypothesis testing (whether condition and age have a significant effect on children's success) rather than about an exploratory approach to finding the "best" model. More importantly, however, I would like to draw the authors' attention to a publication by Mundry and Nunn (2009; and references herein) amplifying "previous warnings about using stepwise procedures" (p. 119) concluding that these procedures "should not be used in the context of testing null hypotheses about a set of predictor variables" (p. 121). This is because stepwise methods lead to greatly inflated type 1 error rates due to multiple testing. I therefore suggest removing the stepwise selection procedure from the data analysis.

Studies using GLMMs usually first conduct a test of a full model against a null model (i.e., a model containing all control variables and random effects but lacking the crucial predictor variables (which in the current study would be age and condition (and trial number)) and only if this analysis reveals a significant result, effects of individual predictors should be analysed (see Forstmeier & Schielzeth, 2011; for examples of such approaches see Göckeritz, Schmidt, & Tomasello, 2014; Kalan, Mundry, & Boesch, 2015; Völter, Sentís, & Call, 2016). I therefore suggest running and reporting a test of the full model before presenting the analysis for the individual predictors.
**Thank you for this useful suggestion, we have now incorporated it within our analysis in place of the stepwise selection process.

**Results**

1. Line 295: In the second sentence of the results section (lines 293-295), it is stated that conditions 2 and 5 differ significantly from the other conditions, and the reader gets referred to Table 1. However, Table 1 does not list the results of the comparisons of the conditions against each other, only the estimate for condition overall. Therefore, I recommend that these missing details (estimates on the individual conditions compared against each other) should be added to Table 1

** This is no longer applicable following new GLMM analysis based on reviewer suggestion.

2. Line 299: Similar to point 1, where the text explains that condition 5 is significantly different from the other conditions, the reader is referred to Table 3 – again, this information cannot be found in the table and needs to be added

**This is now provided in **Supplementary Table 1**.

3. Lines 301/302: More information should be provided with regard to how the analysis of correct choices within each condition per age group was performed. So far, the information that binomial tests were used can only be found in the caption of Table 3. I would recommend to provide this information in the results section (possibly also in the data analysis section) as well.

**This information has been added to **line 334**.

4. Related to point 3, it would be good if the authors could explain why they chose to use binomial tests for this analysis. I wonder whether the same or a similar analysis could have been included into the GLMM (maybe via an interaction) and maybe this would have avoided multiple testing and the need for a Bonferroni correction. However, maybe the sample size would have been too small for including an interaction term into the GLMM. It would be great if the authors could provide more information on why they decided to use this analysis (I do not mean to imply that it is wrong).

** Following your suggestion, we incorporated an interaction term between age and condition in the GLMM analysis, and found this interaction to be significant across 5 trials (Table 1). We then selected binomial tests for further analysis using age groups, as these were generally the statistical methods utilised in previous Aesop's Fable experiments (e.g. Miller et al., 2016 – child and crow Aesop's Fable study, Logan et al., 2014 and Jelbert et al., 2014 crow Aesop's Fable studies), allowing for easy comparison between studies.

**Validity of the findings**

I would like the authors to elaborate on the potential criticism that participants in their study were still able to solve learn the task via visual-motor feedback. One could argue that even though children did not see the water level rising, they do see the "new" water level when they use the fishing rod (at least the girl in the example video provided is looking down the tube). It seems that by being able to see the water level after the objects have been dropped, children are still able to receive visual feedback from the task. Even though they do not actually see the water level rising, they might be able to mentally "reverse-engineer" what was happening when they see the new water level. Another point could be that when children see that the stones that are too large get stuck in the tube this could also provide some kind of perceptual-motor feedback for the children. I think the discussion section would benefit from including this point: In how far might children still have been able to learn via perceptual-motor feedback?

**We have added this to the discussion in **lines 496-504**. Where, importantly, we now highlight the distinction between the perceptual-motor feedback hypothesis, and learning from visual feedback of any kind.

Somewhat related to point 1: In lines 324-326, the authors state "Over five trials, children received feedback about the success of their actions (whether or not the token could now be reached using the fishing rod), but at no point did they observe the water level rising." I would like the authors to elaborate somewhat on this statement. What does it mean that children were able to learn from their actions? Could that be seen as a form of learning via perceptual-motor feedback as well? For example, as indicated above, when children see the objects that are too large getting stuck in the tube, is that not an example of perceptual-motor feedback? Where exactly is the difference between learning from the success of one's actions and learning from perceptual-motor feedback? I find this especially important to answer because in the current study even if the tubes would not have been opaque, due to the fact that all objects were dropped at once learning from one's actions (successfully solved task: yes or no) and learning from perceptual feedback from the rising water level (water level risen: yes or no) coincide (whereas in previous studies because objects had to be dropped repeatedly, feedback from the rising water level and feedback about the success of one's actions was somewhat independent).

** We have elaborated on the sentence highlighted by the reviewer. We now state: 'Over five trials, here, children received feedback about the success of their actions at the end of each trial (whether or not the token could now be reached using the fishing rod), but at no point did they observe the water level rising. This contrasts with previous studies in which subjects were able to observe the change in the water level, within a trial, each time an object was dropped into a tube.' (**lines 406-411**) We have also made this distinction clearer in the introduction (**lines 134 onwards**).

Lines 399-400: Given the above point with regard to whether it could still be possible that children were able to learn how to solve these tasks via perceptual-motor feedback, the sentence in lines 399-400 might have to be rephrased slightly. I would have written "Overall, the results reported here suggest that young children's success on the first trial of Aesop's Fable tasks cannot be attributed to learning from perceptual-motor feedback." I think this is a conclusion that is clearly supported by the current findings, but I am less convinced that the current study ruled out the possibility that children's learning took place without visual-motor feedback of some sort.

**We have rephrased the paragraph that includes this sentence (**lines 490 onwards**) to make it clear that ruling out the perceptual-motor feedback hypothesis is distinct from ruling out the use of visual feedback of any kind.

Line 405: It is great that the authors draw a connection to literature from a different field (innovative abilities in children) but I think another sentence would be needed to explain why this literature was mentioned. What do these studies and the current study have in common? Do they hint at the same cognitive processes involved?

**We have added an additional sentence to clarify what these studies have in common (**line 507**).

Regarding the discussion of the "narrow vs wide tube" condition (lines 388-398): When reading through the methods and looking at Fig. 1, I found the solution in the "narrow vs wide tube" condition not as intuitive as the solutions in the other tasks. I think it might be premature to

conclude that children's poor performance in this task indicates a lack of understanding ("are not yet able to accurately simulate the different effects that objects will have on the water level of differently sized tubes", lines 393-395). It might be worthwhile to consider testing this condition with a small sample of adults to see how they perform on this task (not necessarily for this revision, but it should be considered for the future). If children's poor performance is actually due to a lack of understanding, one would expect adults to show excellent performance on first trial, but if adults struggle as well, one might have to question the validity of the task and to adjust it accordingly – the authors themselves already acknowledged that the difference between the two tubes was only subtle (line 387).

**We thank the reviewer for the suggestion that we test the narrow and wide tube condition with an older sample to determine whether it is possible to solve on the first trial. We now highlight that children struggle 'in this context', and remove the word 'yet', to limit our conclusions to this specific task rather than making a more general conclusion here (**line 486**).

Line 365: It could be stated a bit more clearly what the procedure in Miller et al 2016 was. Specifically, it could be stated explicitly that children were allowed to select several types of objects – in the current manuscript, this information has to be inferred from line 365. So maybe state in half a sentence that children could insert both hollow and solid objects into the tubes.

** Added as requested (**line 451**).

**Reviewer 2**
**Basic reporting**
Figures 2 and 3 need information in the legend to explain what the error bars are (I assume they are standard error bars, but it is not clear). Table 3 should state what the adjusted p-value threshold is following Bonferroni correction. Column headings are also inconsistently labelled as both upper and lowercase 'p'.

**All changes made as requested.

Where citations that cover a range of different species are mentioned, it would be helpful to identify which paper tested which species (e.g. the list given lines 71-72), as the authors do elsewhere (e.g. lines 90-93).

**Changed as requested.

**Experimental design**
1. Each of the tubes includes a clear section at the bottom, even when the opaque part of the tube is rotated. This is evident from Figure 1, and from the supplementary video. Since the top part of the tube is covered, children cannot see the water level rising (thus preventing access the main source of perceptual-motor feedback), but they can see (for example) the sinking objects appearing at the bottom, in contrast to floating objects. Can the authors explain the rationale behind leaving a portion of the tube uncovered at the bottom?

**This was to ensure the children could see that the objects had rolled into the tube and not elsewhere, but the object's effects on the water level in the tube were obscured (**line 295**).

2. A first reading of lines 195-197 makes it appear that at least one of the conditions might be split across days, because 18 trials are completed in one day, and there are (apparently) 5 trials per condition. It is made clear that this is not the case in line 254, because - as far as I understand it - each 'block' is composed of 1 trial of each of the 6 conditions, intermixed. Thus an 18/12 split of trials allows for complete sets of these 6 trial blocks on each day. It would be helpful to explain the trial structure earlier to avoid this confusion. Also, more detail on the pseudorandom scheme used would be helpful.

**We now mention the randomisation procedure in **line 236**, and state how this spanned the two testing days. We have clarified our description of the pseudo-randomisation in **lines 300-305**.

3. It is not immediately clear whether the functional distinction between the hollow and solid cubes in Condition 4 is obvious from observing the objects. Looking closely at the cubes depicted in Figure 1D, it appears that the 'hollow' object is actually more like a wireframe cube, without solid sides, rather than the solid-sided but hollow interior object I had imagined. Altering the description in the text would avoid this confusion.

**We now state that the hollow cubes comprise a hollow wire frame, and direct the reader to **Figure 1**.

4. Are the ages given in line 147 school year ages or actual ages? It would be helpful to state the actual mean ages (plus range) in years and months, since there can be substantial variation within a school year group.

**We report actual ages rather than school years, and have provided the mean and range for each age group in years in the methods section.

5. Line 281 ("Each dropped variable was then re-added separately to the final model to check it remained significant") is confusing. Perhaps the authors could elaborate slightly on the model selection process.

**Following the suggestion of another reviewer, we have conducted the GLMM's using a different method than the step-wise selection procedure. "We conducted Generalized Linear Mixed Models (GLMM: Baayen, 2008) using R (version 2.15.0; R-Development-Core-Team, 2012) to assess which factors influenced success rate in the children. Success was a binary variable of whether the subject correctly solved the trial (1) or not (0), and was entered as a dependent variable in the models. We ran two models as we had two measures of interest: model 1) success on the first trial and model 2) success across all five trials. We included the random effect of subject ID, fixed effects of age in years (ages 5-9 in individual years), condition (1-6), gender (male/female), trial number (1-5; model 2 only) and the interaction between age and condition, and age and trial number (model 2 only). We used likelihood ratio tests to compare the full model (all predictor variables, random effects and control variables) firstly with a null model, and then with reduced models to test each of the effects of interest (Forstmeier & Schielzeth, 2011). The null model comprised of random effects, control variables and no predictor variables. The reduced model comprised of all effects present in the full model, except the effect of interest (Gockeritz, Schmidt & Tomasello, 2014)."

**Validity of the findings**
The authors should provide a rationale for grouping children into two age categories (5-7 and 8-9) for the analysis of choices within each condition (lines 301-315, Table 3), and specifically why this particular grouping was made rather than analysing each age separately.

** Children were grouped into 5-7, and 8-9 years, rather than as single year groups, to allow for comparisons with previous studies, while reducing the possibility of Type II errors due to small sample sizes. We have added this to **line 336**. However, for the interested reader we now also include this analysis per year group in the supplementary materials (**Table S2**)

Finally, the authors use the phrase "mentally simulate" several times to describe the putative cognitive mechanism that children (or other species) solving the task on the first trial without perceptual-motor feedback might be using (e.g. in the abstract, lines 82, 323, 354). I suspect that this is the most likely explanation, but since other mechanisms are possible, and the authors did not ask children to describe how they solved the task, I think this term is too specific. If used, it should be acknowledged that there are other potential mechanisms that would fit the observed results.
**We have rephrased our use of the term mentally simulate (e.g. in the abstract, **line 404**) so that we no longer imply that this is certainly the mechanism being used.

---

## Round 0.3 · accepted · Accept

· Academic Editor

Accept

Thank you for attending to these final, minor issues, and thanks again for submitting such a nice paper to PeerJ.